# Ceramic-like stable CsPbBr$_3$ nanocrystals encapsulated in silica derived from molecular sieve templates

Qinggang Zhang [1,3], Bo Wang[1,3], Weilin Zheng[1], Long Kong [1], Qun Wan[1], Congyang Zhang[1], Zhichun Li[1], Xueyan Cao[1], Mingming Liu[1] & Liang Li [1,2]*

Achieving good stability while maintaining excellent properties is one of the main challenges for enhancing the competitiveness of luminescent perovskite CsPbX$_3$ (X=Cl, Br, I) nano-crystals (NCs). Here, we propose a facile strategy to synthesize ceramic-like stable and highly luminescent CsPbBr$_3$ NCs by encapsulating them into silica derived from molecular sieve templates at high temperature (600–900 ºC). The obtained CsPbBr$_3$-SiO$_2$ powders not only show high photoluminescence quantum yield (~71%), but also show an exceptional stability comparable to the ceramic Sr$_2$SiO$_4$:Eu$^{2+}$ green phosphor. They can maintain 100% of their photoluminescence value under illumination on blue light-emitting diodes (LEDs) chips (20 mA, 2.7 V) for 1000 h, and can also survive in a harsh hydrochloric acid aqueous solution (1 M) for 50 days. We believe that the above robust stabilities will significantly enhance the potential of perovskite CsPbX$_3$ NCs to be practically applied in LEDs and backlight displays.

[1] School of Environmental Science and Engineering, Shanghai Jiao Tong University, 800 Dongchuan Road, Shanghai 200240, China. [2] Shanghai Institute of Pollution Control and Ecological Security, Shanghai 200092, P. R. China. [3] These authors contributed equally: Qinggang Zhang, Bo Wang *email: liangli117@sjtu.edu.cn

Light-emitting diodes (LEDs) have been successfully used in lighting and liquid crystal displays due to their tunable color, high efficiency, long lifetime, durability, and energy saving[1–3]. The majority of current commercial phosphor-converted LEDs can be achieved by the combination of a blue InGaN chip with single or multiple phosphors. For the display applications, the narrow emission width of phosphors is a critical factor to achieve wide display's color gamut without sacrificing power efficiency[4]. In this respect, red or green phosphors with a narrow band emission have attracted a lot of attention, such as $Sr_2MgAl_{22}O_{36}$: $Mn^{2+}$, $Ba_2LiSi_7AlN_{12}.Eu^{2+}$, $K_2SiF_6:Mn^{4+}$ (KSF), and $SrLiAl_3N_4$: $Eu^{2+}$[5,6]. Moreover, quantum dots (QDs), i.e. semiconductor nanocrystals (NCs) including II–VI, III–V, and perovskite NCs, have also emerged as highly credible options for displays thanks to their narrow emissions and design flexibilities (adjustable emission wavelengths)[7,8]. However, practical display applications not only strive for high efficiency and wide color gamut, but also for cost-competitiveness and operational stability. From the cost considerations, perovskite NCs could be one of the most cost-competitive down-conversion emitters for display applications due to their easy synthesis and cheap raw materials. But from the stability perspective, perovskite NCs including $CsPbX_3$ (X = Cl, Br, I) NCs may be one of the worst types because their operational stability is far inferior to the ceramic phosphors, which is even worse than the conventional II–VI and III–V QDs, owing to their intrinsically moisture/heat/light sensitive ionic structures[9]. Obviously, achieving good stability while maintaining their excellent properties is one of the main challenges for the practical applications of perovskite $CsPbX_3$ NCs.

So far, various strategies have been developed to stabilize $CsPbX_3$ NCs[9]. A routine strategy is to coat the NCs with inert shells or incorporate them into barrier matrixes, which can isolate $CsPbX_3$ NCs from moisture and oxygen, and also prevent ion migration and the induced inter-particles fusing[9]. For example, the stabilities of $CsPbX_3$ NCs have been improved by encapsulating them into inorganic oxides ($SiO_2$[10], $Al_2O_3$[11], $SiO_2/Al_2O_3$[12], $TiO_2$[13], $ZrO_2$[14]), mesoporous materials (mesoporous silica[15], metal−organic frameworks[16]), polymer matrixes (polystyrene[17], polymethyl methacrylate[18], polyvinylidene fluoride[19]), inorganic salts ($NaNO_3$[20], $NH_4Br$[21]), and shell formation ($CsPbBr_3/CsPb_2Br_5$[22], $CsPbBr_3/Cs_4PbBr_6$[23], $CsPbBr_3/Rb_4PbBr_6$[24]). However, these shells or barrier matrixes can only slow down the degradation of $CsPbX_3$ NCs by the external environmental factors, and their stability is still much worse than the ceramic phosphors. Generally, the failure of the protective strategy could be mainly attributed to the following three reasons: (1) the shell or matrix materials cannot completely protect $CsPbX_3$ NCs, such as the porous matrixes, in which the pore structures are exposed, and cannot completely isolate perovskite NCs from moisture and oxygen; (2) the shell or matrix materials are not intrinsically stable, such as inorganic salts ($NaNO_3$, $NH_4Br$, $CsPb_2Br_5$, $Rb_4PbBr_6$) which are still sensitive to moisture and oxygen; (3) the shell or matrix materials are stable and can completely coat on $CsPbX_3$ NCs, such as inorganic oxides ($SiO_2$, $Al_2O_3$, $SiO_2/Al_2O_3$, $TiO_2$, $ZrO_2$), but are not dense enough and still have some morphological pinholes, which cause the high permeable rates of external $H_2O/O_2$. Actually, these inorganic oxides require high synthesis or annealing temperature in order to achieve dense oxides with few pinholes and great barrier properties, since their densification extent is strongly dependent on the annealing temperature. Some studies suggest that high annealing temperatures above 800 °C could promote the transition from amorphous to crystalline and get much better barrier property for $SiO_2$ and $Al_2O_3$ thin film[25,26]. A big challenge for $CsPbX_3$ NCs is that they cannot withstand such a high temperature. In our previous report, the annealing temperature of $CsPbBr_3/SiO_2/Al_2O_3$ could

not exceed 150 °C[12] due to the severe surface oxidations or fusing of $CsPbX_3$ NCs. The organic ligands on the surface of $CsPbX_3$ NCs will be oxidized (exceed 150 °C), and higher temperatures may damage or peel off the organic ligands and accelerate ion migration of inter-particles, thus $CsPbX_3$ NCs will be agglomerated that lead to the fluorescence quenching. It is difficult to encapsulate perovskite NCs with dense inorganic oxides at high temperature and simultaneously keep their morphology and optoelectronic properties unchanged.

Here, we propose a facile strategy to synthesize ceramic-like stable and highly luminescent $CsPbBr_3$ NCs by template confined solid-state synthesis and in situ encapsulation, which is based on a strategical collapse of the silicon molecular sieve (MS) template at a high synthesis temperature (600–900 °C). The synthesis process is a solid-state reaction at high temperature without organic solvents and organic ligands. The collapsed MS not only confine the growth of $CsPbBr_3$ NCs, but also block the interaction of $CsPbBr_3$ NCs at high temperature. The as prepared $CsPbBr_3$–$SiO_2$ micron-size powders not only show high photoluminescence quantum yield (PLQY) up to 71% and a narrow emission with a full width at half maximum (FWHM) around 20 nm, but also show exceptional photostabilities even slightly better than the ceramic $Sr_2SiO_4:Eu^{2+}$ green phosphor under our testing conditions. Furthermore, thanks to the complete encapsulation of dense $SiO_2$ at high temperature, the as prepared $CsPbBr_3$-$SiO_2$ powders can survive in a harsh hydrochloric acid aqueous solution (1 M HCl) for 50 days without obvious photoluminescence (PL) intensity changing, which prove the excellent barrier properties of the formed $SiO_2$ solid. Even the small $Cl^-$ ions cannot pass through the $SiO_2$ protective layer to reach $CsPbBr_3$ NCs. We believe the $CsPbBr_3$–$SiO_2$ powders have tremendous potential for LEDs and backlight display applications based on their excellent optical properties and stability.

## Results

**Synthesis and encapsulation of $CsPbBr_3$ into MS.** All-silicon molecular sieves (MCM-41) were chosen as the MS template (Supplementary Table 1) because of their large surface area, narrow pore size distribution (d = 3.6 nm). More importantly, their pore structures can collapse at certain high temperature[27]. As illustrated in Fig. 1a, the porous MS template was firstly soaked into the precursor salts (CsBr and $PbBr_2$) solution, and then dried at 80 °C. The resultant mixture was placed into a furnace and heated up to a high temperature (600–900 °C). At this high temperature, $CsPbBr_3$ NCs were synthesized in the confined pores of MS, meanwhile, the pore structures of MS gradually collapsed and encapsulated the $CsPbBr_3$ NCs, finally forming a dense $CsPbBr_3$–$SiO_2$ solid.

Figure 1b illustrated the colors of $CsPbBr_3$–$SiO_2$ (mass ratio of $CsPbBr_3$: MS is 1:3) which gradually changed from white to deeper yellow with the synthesis temperature increasing from 400 °C to 700 °C, and then turned into lighter colors when the temperature changed to 800 °C and 900 °C. The XRD patterns (Fig. 1c) confirmed the successful formation of cubic $CsPbBr_3$ NCs (PDF# 54-0752) when the reaction temperature was above 500 °C. In addition to the diffraction peaks from the cubic $CsPbBr_3$ NCs, a broad shoulder around 23° C in $CsPbBr_3$–$SiO_2$ powders was also observed, which belonged to the amorphous phase of $SiO_2$ (Fig. 1c and Supplementary Fig. 1a). For the sample calcined at 400 °C ($CsPbBr_3$-$SiO_2$–400), except the existence of the cubic $CsPbBr_3$ structure, we also observed the sharp diffractions from $CsPb_2Br_5$ (PDF# 25-0211) and CsBr (PDF# 52-1144, red arrows), but the signal from the latter structure was weak, which was similar to the dried raw mixture (Supplementary Fig. 1b) without calcination, indicating that there was no obvious

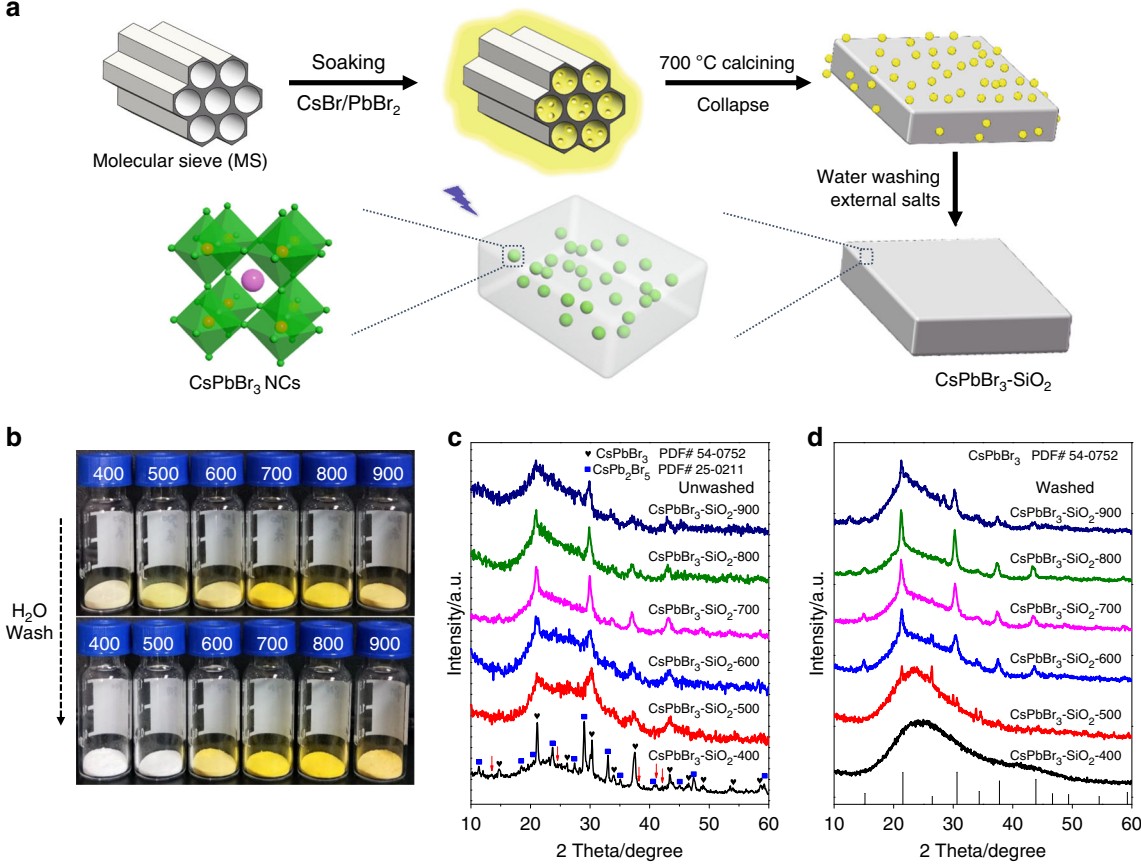

**Fig. 1 Growth and encapsulation of CsPbBr₃ NCs into MS matrixes at high temperature. a** The schematic diagram of synthesis CsPbBr₃ NCs into MS (SiO₂). **b** Photographs of the unwashed CsPbBr₃–SiO₂ powders (upper) and water washed CsPbBr₃–SiO₂ powders (bottom) at different calcination temperature under visible illumination, CsBr/PbBr₂:MS = 1:3. **c** XRD patterns of unwashed CsPbBr₃–SiO₂ powders. **d** XRD patterns of water washed CsPbBr₃-SiO₂ powders.

chemical reaction at 400 °C. These large CsPbBr₃ and CsPb₂Br₅ crystals were formed in the dry process (Eqs. 1–2)[28]:

$$2PbBr_2 + CsBr \rightarrow CsPb_2Br_5 \qquad (1)$$

$$CsPb_2Br_5 + CsBr \rightarrow 2CsPbBr_3 \qquad (2)$$

Upon increasing the calcination temperature at 500 °C, the diffraction peaks from CsPb₂Br₅ structure disappeared gradually and the peaks from cubic CsPbBr₃ became more obvious because of the decomposition of CsPb₂Br₅ into CsPbBr₃ (Eq. 3)[29]. It is also possible that the unreacted CsBr was converted into CsPbBr₃ (Eq. 4).

$$CsPb_2Br_5 \rightarrow CsPbBr_3 + PbBr_2 \qquad (3)$$

$$PbBr_2 + CsBr \rightarrow CsPbBr_3 \qquad (4)$$

The melting point of CsPbBr₃ is about 567 °C, and its sublimation starts when it is melted[30,31]. Therefore, when the temperature was heated up to 600 °C, the large CsPbBr₃ crystals started melting and sublimating, which filled into the pores of MS and formed CsPbBr₃ NCs when the temperature was cooling down. Simultaneously, the pores of MS started to collapse and finally fused into a dense SiO₂ solid at high temperatures. The sealed SiO₂ nanopore not only confined the growth of CsPbBr₃ NCs, but also protected the formed CsPbBr₃ NCs from oxidation and inter-particles fusion at high temperature.

To see if the CsPbBr₃ NCs are really incorporated into the SiO₂ solid, the samples were washed with water. Figure 1d showed the XRD patterns of CsPbBr₃–SiO₂ powders after water washing, and

only the samples with higher calcination temperature (≥600 °C) maintained the diffractions features of cubic CsPbBr₃ NCs. In contrast, the lower temperature synthesized samples (CsPbBr₃–SiO₂-400, CsPbBr₃–SiO₂-500) almost lost all the diffraction peaks after water washing, showing that the perovskite crystals were not sealed yet. Supplementary Fig. 2 showed the UV-Vis absorption spectra of CsPbBr₃-SiO₂ powders before and after water washing. Except for CsPbBr₃–SiO₂–400, all the other samples presented an absorption band edge around 507 nm before water washing. After water washing, similar to the observation from the XRD patterns, the samples with lower calcination temperatures (400 °C, 500 °C) almost completely lost their absorption because the water damaged or dissolved the unprotected CsPbBr₃ NCs. Obviously, the samples synthesized at higher temperatures had better water resistance. In the following study, CsPbBr₃–SiO₂ meant the samples after water washing if there is no particular explanation.

To explore the formation process of CsPbBr₃ NCs and the evolution of the pore structures of MS at high temperature, TEM images of CsPbBr₃-SiO₂ powders (washed) at different calcination temperatures are shown in Fig. 2a–f. TEM images of CsPbBr₃–SiO₂-400 and CsPbBr₃–SiO₂-500 exhibited no CsPbBr₃ NCs, and similar pore structures as the original MS were still clearly observed (Supplementary Fig. 3), indicating that the pores of MS have not collapsed yet. As the calcination temperature reached 600 °C, the pore structures of MS started to collapse, and some tiny CsPbBr₃ NCs appeared in MS matrixes (Fig. 2c). When the temperature reached 700 °C, the MS obviously became a compact solid, meantime the average sizes

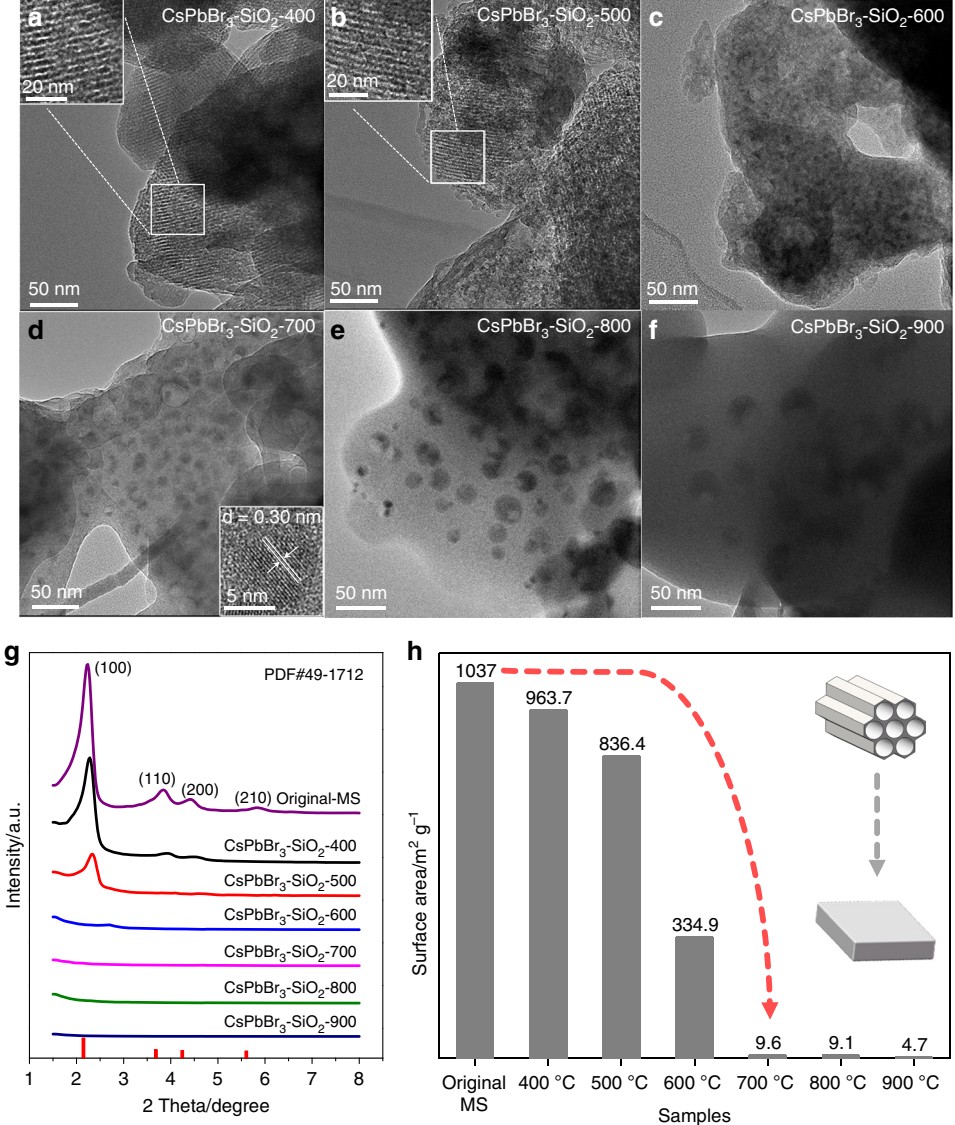

**Fig. 2 The evolution of CsPbBr₃ NCs and the pore structures of MS with temperature increasing.** TEM images of water washed CsPbBr₃–SiO₂ at different calcination temperatures: **a** CsPbBr₃–SiO₂-400, **b** CsPbBr₃–SiO₂-500, **c** CsPbBr₃–SiO₂-600, **d** CsPbBr₃–SiO₂-700, **e** CsPbBr₃–SiO₂-800, **f** CsPbBr₃–SiO₂-900. **g** Small-angle XRD patterns of original MS and CsPbBr₃–SiO₂. **h** Surface area of original MS and CsPbBr₃–SiO₂ (CsBr/PbBr₂: MS = 1:3) calculated with BET method.

of CsPbBr₃ NCs increased gradually with increasing calcination temperature as shown in Fig. 2c–f and Supplementary Fig. 4 (CsPbBr₃–SiO₂-600, d = 6.7 nm; CsPbBr₃–SiO₂–700, d = 9.5 nm; CsPbBr₃–SiO₂–800, d = 20.9 nm; CsPbBr₃–SiO₂-900, d = 30.1 nm). The increased sizes of CsPbBr₃ NCs may be attributed to the softening and collapse of pores of MS with increasing temperature, leading to a weaker template confinement effect, which allowed the growth of larger particles. Before the water washing step, some reaction residues and salts were observed on the surface of unwashed CsPbBr₃-SiO₂–700 as shown in Supplementary Fig. 5a. After water washing, only those fully encapsulated CsPbBr₃ NCs (d = 9.5 nm) were left, and the high-resolution TEM (HRTEM) image clearly exhibited a good crystallinity of CsPbBr₃ NCs with a lattice spacing of 0.30 nm corresponding to the lattice fringes of the (200) planes (Fig. 2d, inset). Compared with the unwashed CsPbBr₃-SiO₂-700, the PL intensity of the water washed CsPbBr₃–SiO₂–700 had been improved due to the removing of reaction residues and salts from

the surface of SiO₂ (Supplementary Figs. 5 and 6), which usually were non-luminescent.

The collapse of the pore structures of MS was also confirmed by small-angle XRD patterns (Fig. 2g). With the increase of calcination temperature, the intensities of the major peaks from the hexagonal structure (PDF# 49-1712) of MS decreased gradually and pore structures were damaged[27]. When the calcination temperature reached 700 ºC, the hexagonal structure of MS completely disappeared. Meanwhile, the surface area of CsPbBr₃–SiO₂ decreased gradually as the temperature increased (Fig. 2h). Compared with the original MS, the surface area of CsPbBr₃–SiO₂–700 decreased from 1037 m² g⁻¹ to 9.6 m² g⁻¹, which indicated that MS had collapsed completely at 700 ºC and most of the CsPbBr₃ NCs were encapsulated into the compact SiO₂ solid during the collapse process. The complete collapse of MS played a key role in protecting CsPbBr₃ NCs from the damages of moisture, because CsPbBr₃ NCs can be completely encapsulated into the dense SiO₂ solid. As we can see, no obvious

PLQYs decays were observed during CsPbBr$_3$–SiO$_2$ (700 °C, 800 °C, and 900 °C) immersed in water for 50 days (Supplementary Fig. 7), but the PLQY of CsPbBr$_3$–SiO$_2$–600 decreased slightly owing to the incomplete collapse of MS at 600 °C.

**Optical characterization of CsPbBr$_3$-SiO$_2$.** The PL spectra of CsPbBr$_3$-SiO$_2$-700 and ceramic Sr$_2$SiO$_4$: Eu$^{2+}$ green phosphor are shown in Fig. 3a. The FWHM of CsPbBr$_3$–SiO$_2$–700 is just 20 nm, which is much narrower than that of ceramic Sr$_2$SiO$_4$: Eu$^{2+}$ green phosphor (FWHM = 62 nm), showing that narrow-band emitting CsPbBr$_3$-SiO$_2$-700 has a great potential to become an outstanding candidate in advanced wide-color-gamut backlight display. Figure 3b showed the PLQYs of CsPbBr$_3$–SiO$_2$ powders synthesized with different experimental conditions. Obviously, CsPbBr$_3$–SiO$_2$–700 (mass ratio of CsPbBr$_3$: MS is 1:3, $t$ = 700 °C) exhibited the highest PLQY of 63% (Fig. 3b and Supplementary Table 3). To better understand the change in PLQYs of CsPbBr$_3$–SiO$_2$ with different calcination temperatures, the PL decay curves of the CsPbBr$_3$–SiO$_2$ were shown in Supplementary Fig. 8 and Supplementary Table 4. The average lifetimes of CsPbBr$_3$-SiO$_2$ gradually increased with the synthesis temperature increasing from 400 °C to 700 °C, and then began to decrease when the temperature reached 800 °C and 900 °C. Particularly, the longest average lifetime was 21.81 ns at 700 °C. The longer lifetimes usually indicated the suppression of the nonradiative decay, and the generated excitons were more inclined to recombine with the radiative path, which was in consistent with the PLQYs with the change of synthesis temperatures. On the other hand, the mass ratios of CsPbBr$_3$: MS (CsBr/PbBr$_2$: MS) profoundly affect the optical properties of CsPbBr$_3$-SiO$_2$ because of the availability of the pores/cavities of MS that may encapsulate CsPbBr$_3$ NCs (Supplementary Figs. 9–11).

**HF etching CsPbBr$_3$–SiO$_2$ for improving the PLQY.** Although the PLQYs of CsPbBr$_3$-SiO$_2$ have been optimized by changing the calcination temperature and the ratio of CsPbBr$_3$: MS, the optimized PLQYs are still limited owing to thick SiO$_2$ covered on the CsPbBr$_3$ NCs, which may hinder the light absorption and conversion of CsPbBr$_3$ NCs. Hence, a HF solution was used to etch SiO$_2$ and remove incomplete encapsulated CsPbBr$_3$ NCs on the surface of CsPbBr$_3$–SiO$_2$, and the treated sample was named as CsPbBr$_3$-SiO$_2$-HF (Fig. 4a). As shown in Fig. 4b and Supplementary Fig. 12, after etching, the CsPbBr$_3$ NCs in the MS seemed more uniform and kept the same lattice spacing of 0.30 nm (Fig. 4b inset). As shown in the high-angle annular dark-field scanning

transmission electron microscopy (HAADF-STEM) image and its corresponding elemental mappings of CsPbBr$_3$–SiO$_2$–HF (Fig. 4c), the Cs, Pb, Br elements were clearly distributed in the CsPbBr$_3$ NCs, indicating that these CsPbBr$_3$ NCs were not destroyed by HF solution because they were deeply buried into SiO$_2$ matrixes. The scanning electron microscopy (SEM) images (Supplementary Fig. 13) showed CsPbBr$_3$–SiO$_2$–700 and CsPbBr$_3$–SiO$_2$–HF are in particle morphology with sizes around 0.5 um~1um. Compared with the smooth unetched sample, the surface of CsPbBr$_3$–SiO$_2$–HF appeared with some holes due to the removal of surface partially-embedded CsPbBr$_3$ NCs, meantime the surface area of CsPbBr$_3$-SiO$_2$-HF increased from 9.6 m$^2$ g$^{-1}$ to 16.7 m$^2$ g$^{-1}$ (Supplementary Table 5). As expected, both PL peak position and absorption edge of CsPbBr$_3$-SiO$_2$-HF blue-shifted around 2 nm owing to the removal of larger partially-embedded CsPbBr$_3$ NCs after HF etching (Fig. 4d). Therefore, CsPbBr$_3$–SiO$_2$–HF showed a brighter green fluorescence (Fig. 4e) with improved absolute PLQY from 63% to 71% (Supplementary Table 3) without any changes in XRD patterns (Fig. 4f). The PL decay curves of the CsPbBr$_3$–SiO$_2$–700 and CsPbBr$_3$–SiO$_2$–HF were shown in Supplementary Fig. 14 and Supplementary Table 6. There was no significant difference in average fluorescence lifetimes between CsPbBr$_3$–SiO$_2$–700 ($\tau$ = 21.81 ns) and CsPbBr$_3$–SiO$_2$–HF ($\tau$ = 22.14 ns). The HF etching did not change the structures and properties of CsPbBr$_3$ NCs. In other words, the improved PLQY was mainly attributed to the thinning of SiO$_2$ shell.

**Water stability, chemical stability, and photostability.** As is known to all, the stability of CsPbBr$_3$ NCs is challenged by ambient factors such as light irradiation, water, oxygen, and heat, which restrict their applications[9]. In this work, CsPbBr$_3$ NCs were completely encapsulated into the high temperature annealed SiO$_2$ solid, and the water washing step already proved the good water-resistance of the as prepared CsPbBr$_3$-SiO$_2$ powders. To further quantify their water-resistant capability, CsPbBr$_3$–SiO$_2$–700 and CsPbBr$_3$–SiO$_2$–HF were dispersed into water with PLQY monitoring. For comparison, ceramic Sr$_2$SiO$_4$:Eu$^{2+}$ green phosphor (Intermtix.co), KSF red phosphor (Intermtix.co), and colloidal CsPbBr$_3$ NCs were used as references. As shown in Fig. 5a, immediate degradation of KSF red phosphor and colloidal CsPbBr$_3$ NCs were observed in one hour. As contrast, Sr$_2$SiO$_4$: Eu$^{2+}$ phosphor, CsPbBr$_3$–SiO$_2$–700, and CsPbBr$_3$–SiO$_2$–HF were dispersed in water for 50 days and still exhibited bright green fluorescence (Supplementary Figs. 15–16a). No obvious PLQYs reduction was observed in CsPbBr$_3$–SiO$_2$–700 and CsPbBr$_3$–SiO$_2$–HF (Fig. 5b), but

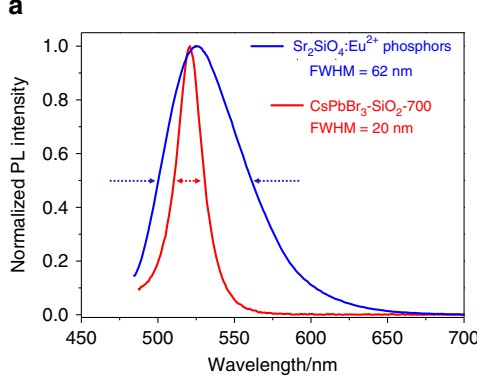

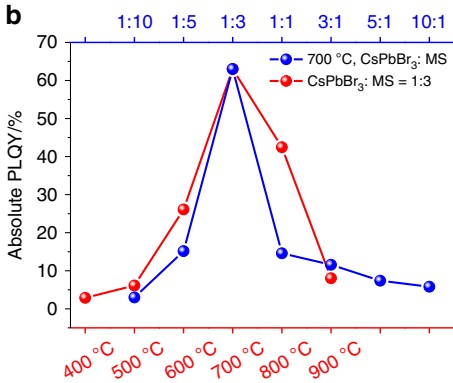

**Fig. 3 Optical properties of CsPbBr$_3$-SiO$_2$. a** Photoluminescence emission spectra of CsPbBr$_3$–SiO$_2$–700 and ceramic Sr$_2$SiO$_4$: Eu$^{2+}$ green phosphor, excitation wavelength is 455 nm. **b** Absolute PLQYs of CsPbBr$_3$-SiO$_2$ (mass ratio of CsPbBr$_3$: MS is 1:3, red mark) synthesized at different temperatures, absolute PLQYs of CsPbBr$_3$-SiO$_2$ synthesized at 700 °C with different mass ratios of CsPbBr$_3$: MS (blue mark).

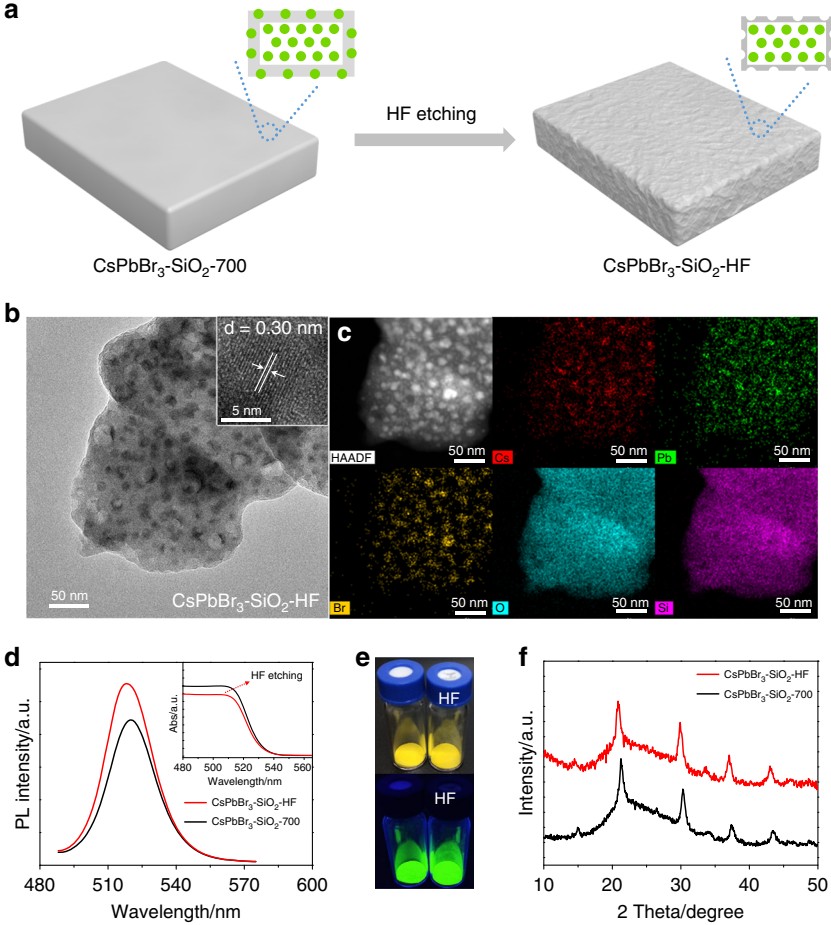

**Fig. 4 Improving PLQYs by HF etching. a** The schematic diagram of CsPbBr₃–SiO₂–700 by HF etching. **b** TEM image of CsPbBr₃–SiO₂–HF. **c** HAADF-STEM image of CsPbBr₃–SiO₂–HF and the corresponding elemental mapping of Cs, Pb, Br, O, and Si. **d** Photoluminescence emission and UV-Vis absorption spectra (insert) of CsPbBr₃–SiO₂–700 and CsPbBr₃–SiO₂–HF. **e** Photographs of the CsPbBr₃–SiO₂–700 powders (left) and CsPbBr₃–SiO₂–HF powders (right) under visible illumination (upper) and UV excitation at 365 nm (bottom). **f** XRD patterns of CsPbBr₃–SiO₂–700 powders, and CsPbBr₃–SiO₂–HF powders.

the PLQYs of $Sr_2SiO_4$:$Eu^{2+}$ phosphor, KSF red phosphor, and colloidal $CsPbBr_3$ NCs decreased to 88%, 17%, and 13% of their initial PLQYs, respectively. More surprisingly, $CsPbBr_3$–$SiO_2$–700 and $CsPbBr_3$–$SiO_2$–HF showed no change even testing in water under strong illumination of a 450 nm LED light (175 mW cm$^{-2}$) for 50 days (Fig. 5c and Supplementary Figs. 15–16b). Except the excellent water resistance and photostability, the samples also showed a robust chemical stability because they could even survive in the strong acid aqueous solution (1 M HCl) for 50 days (Fig. 5c and Supplementary Figs. 15–16c), which indicated that even small $Cl^-$ ions cannot pass through the dense $SiO_2$ shell and cause ion exchange. XRD patterns of the $CsPbBr_3$–$SiO_2$–700 and $CsPbBr_3$–$SiO_2$–HF also remained unchanged after 50 days in the water with light illumination or in 1 M HCl solution (Supplementary Fig. 17), which meant that the dense $SiO_2$ shell can effectively protect $CsPbBr_3$ NCs from water-induced structure collapse, photodegradation, and ion migration.

To verify the potential of $CsPbBr_3$–$SiO_2$ in backlight displays, we tested the operational stability of $CsPbBr_3$–$SiO_2$–HF powders sealed with Norland-61 on the blue LED chips (peak at 455 nm, 20 mA, 2.7 V) under room temperature. As for comparison, colloidal $CsPbBr_3$ NCs, CdSe/CdS/ZnS NCs, ceramic $Sr_2SiO_4$: $Eu^{2+}$ green phosphor, and commercial KSF red phosphor were used as control groups (Supplementary Fig. 18). As shown in Fig. 6a, the relative PL intensity of

$CsPbBr_3$–$SiO_2$–HF still maintained above 100% under illumination for 1000 h, but the relative PL intensities of ceramic $Sr_2SiO_4$: $Eu^{2+}$ green phosphor and commercial KSF red phosphor decreased to 82% and 67% of the initial intensity after 1000 h. Meanwhile, the relative PL intensity of CdSe/CdS/ZnS NCs dropped to 38% after 360 h under illumination and the relative PL intensity of colloidal $CsPbBr_3$ NCs sharply dropped to 15% after 40 h. It was confirmed that $CsPbBr_3$–$SiO_2$–HF exhibited comparable operation stability as the commercial ceramic phosphors. To further prove the temperature and moisture resistance of the $CsPbBr_3$–$SiO_2$ in practical display applications, we performed the accelerated operational stability tests for the above device samples under high temperature (HT 85 °C) and high humidity (HH 85%) conditions (Fig. 6b). After aging for 168 h, the PL intensity of $CsPbBr_3$–$SiO_2$–HF retained almost unchanged, which is more superior to ceramic $Sr_2SiO_4$: $Eu^{2+}$ green phosphor that only remained 80% of the initial PL intensity. As a contrast, the PL intensities of commercial KSF red phosphor, CdSe/CdS/ZnS NCs and colloidal $CsPbBr_3$ NCs sharply decreased to 43% (aging for 168 h), 60% (aging for 20 h), 7% (aging for 20 h) of the initial intensities, respectively. These significant differences undoubtedly proved that the facile strategy for fully encapsulating $CsPbBr_3$ NCs into the dense $SiO_2$ solid can effectively protect $CsPbBr_3$ NCs against the damages from oxygen, moisture, light irradiation, and heat. To our knowledge, the

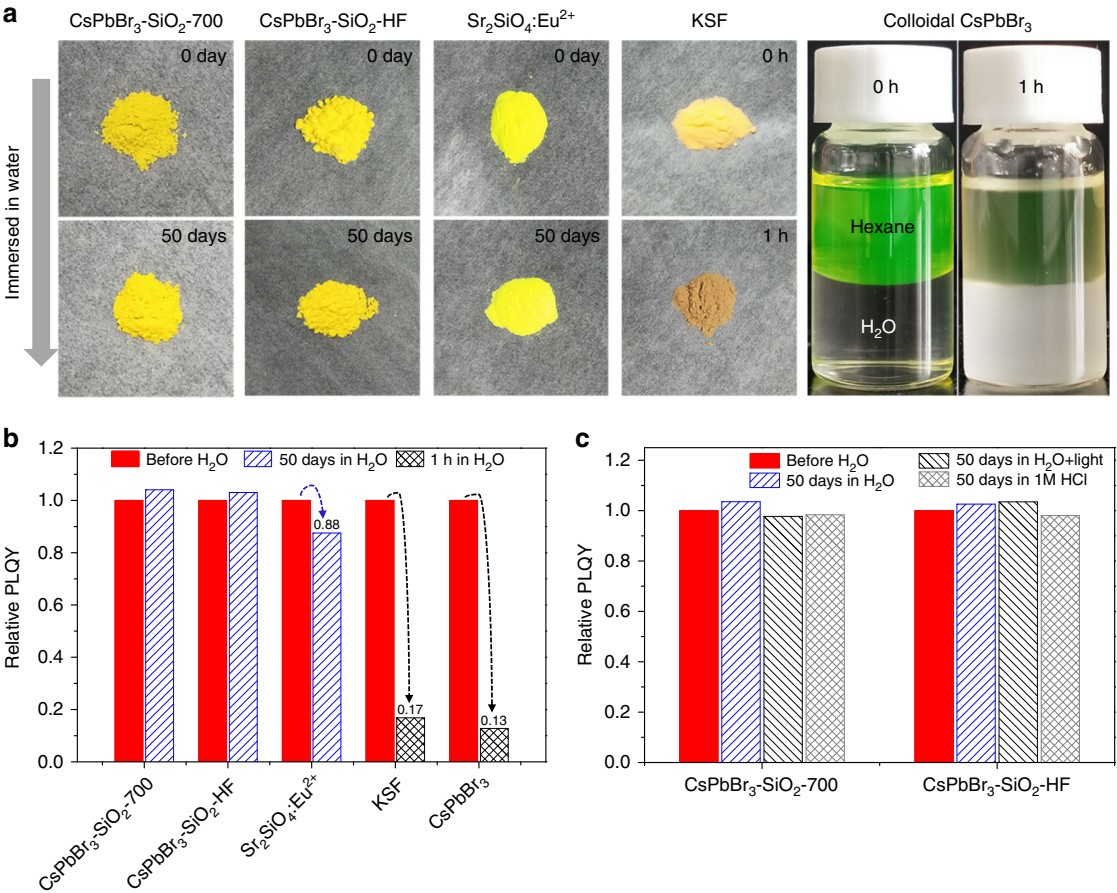

**Fig. 5 Water resistance and acid resistance of CsPbBr₃-SiO₂.** Photographs (**a**) and relative PLQYs (**b**) of the CsPbBr₃–SiO₂–700, CsPbBr₃–SiO₂–HF, ceramic $Sr_2SiO_4$:$Eu^{2+}$ green phosphor, KSF red phosphor, and colloidal CsPbBr₃ NCs after immersed in water for various times. **c** Relative PLQYs of the CsPbBr₃–SiO₂–700 and CsPbBr₃–SiO₂–HF after immersed in various solvents for 50 days, extra light source: a 450 nm LED light (175 mW cm⁻²).

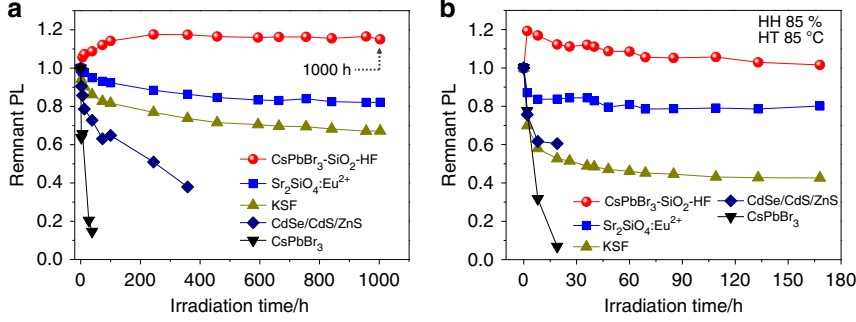

**Fig. 6 Photostability of the CsPbBr₃–SiO₂–HF. a** Photostabilities of the CsPbBr₃–SiO₂–HF, ceramic $Sr_2SiO_4$: $Eu^{2+}$ green phosphor, KSF red phosphor, colloidal CsPbBr₃ NCs and CdSe/CdS/ZnS NCs under illumination, sealed with Norland-61 on the LED chips (20 mA, 2.7 V) and **b** aged at 85 °C and 85% humidity conditions on the LED chips (20 mA, 2.7 V).

stabilities (such as, water stability, chemical stability, and photostability) of CsPbBr₃-SiO₂ are much superior to the reported results of conventional perovskite composites that protected by different coating materials and methods (Supplementary Table 7). Next, to verify the universality of this method, CsPbBr₃ NCs confined in different types of molecular sieves (such as ZSM, NaY, and Y-Zeolite, the pore size distribution from 0.5 nm to 3.6 nm) were further explored (Supplementary Table 1 and Supplementary Fig. 19). As we can see, luminescent CsPbBr₃ NCs can be synthesized in different MS templates with a wide pore size range.

## Discussion

In summary, we have introduced a facile approach for the in situ growth and encapsulation of CsPbBr₃ NCs into SiO₂ at high temperature for improving stability. Based on the specific porous structure of molecular sieves (MCM-41), we were able to synthesize CsPbBr₃ NCs by a nano-confined growth at high temperatures. By smartly applying the specific collapse behavior of MCM-41 at high temperature, we successfully encapsulated the CsPbBr₃ NCs into the dense SiO₂ solid, which offered ceramic-like stability to CsPbBr₃ NCs. Particularly, the PL intensity of CsPbBr₃–SiO₂ remained 100% of its initial value under

illumination on blue LED chips (20 mA, 2.7 V) for 1000 h, even better than the ceramic silicate phosphor. The robust stability, ultra-narrow emission, and high PLQY make the CsPbBr$_3$–SiO$_2$ powders an ideal active material for many optoelectronic applications, particularly as down-conversion emitters for wide color gamut display. Their excellent water/acid-resistance will extend the applications of perovskite NCs, such as in vitro bioimaging/biosensing fluorescent labels even in an acid aqueous medium (stomach), or allow us to perform long term in-vivo tracking of the labeled target if we can decrease the size of CsPbBr$_3$–SiO$_2$ particles to nanoscale.

## Methods

**Chemicals**. Cesium bromide (CsBr, 99.5%), lead bromide (PbBr$_2$, 99%), Cesium carbonate (Cs$_2$CO$_3$, 99.9%), 1-octadecene (ODE, 90%), oleylamine (OAm, 90%) were purchased from Aladdin. Oleic acid (OA, 90%) was purchased from Aldrich. Methyl acetate (98%), toluene (99.5%) were purchased from Sinopharm Chemical Reagent. Molecular sieves (MS) were purchased from Tianjin Yuanli Chemical Co., Ltd. All the chemicals were used without further purification.

**Preparation of CsPbBr$_3$–SiO$_2$**. Briefly, CsBr and PbBr$_2$ (the mole ratio was 1:1) were dissolved into 50 mL ultrapure water in the 250 mL beaker and stirred constantly for 30 min at 80 °C. Then, a certain amount of MS (the mass ratio of CsBr/PbBr$_2$: MS = 1:3) was added into the above solution and the mixture was stirred for 1 h. The as-obtained mixture was dried at 80°C. The collected mixture was ground and calcined at the set temperature for 0.5 h with a heating rate of 5 °C min$^{-1}$ in the muffle furnace under an air atmosphere. After cooling to room temperature, the sample was ground and washed with ultrapure water for several times to remove external CsPbBr$_3$ or other salts. Finally, the washed-sample was obtained by centrifugation and drying at 60 °C. The products obtained at 400°C, 500°C, 600°C, 700°C, 800°C, and 900 °C were denoted as CsPbBr$_3$–SiO$_2$–400, CsPbBr$_3$–SiO$_2$–500, CsPbBr$_3$–SiO$_2$–600, CsPbBr$_3$–SiO$_2$–700, CsPbBr$_3$–SiO$_2$–800, and CsPbBr$_3$–SiO$_2$–900, respectively. The nominal compositions of control groups were shown in Supplementary Table 2.

**HF etching CsPbBr$_3$–SiO$_2$**. 50 mg of CsPbBr$_3$-SiO$_2$-700 was added into 15 mL HF (c = 0.04 M) solution and stirred for 0.5 h. The obtained product was then centrifuged and washed several times with ultrapure water. Finally, the product was dried overnight at 60 ºC. The obtained product was denoted as CsPbBr$_3$–SiO$_2$–HF.

**LED package**. The used UV-cured optical adhesive was Norland-61. In brief, 10 mg of CsPbBr$_3$–SiO$_2$ was mixed with 200 mg of UV-cured optical adhesive. To remove the bubbles from the optical adhesive, the resulting mixture was heated at 40 °C for 0.5 h under vacuum. After that, the mixture was deposited on a 455 nm InGaN LED chip and then UV cured for 50 s (365 nm, 80 W cm$^{-2}$). Then the green CsPbBr$_3$–SiO$_2$ LED was obtained. According to the above similar procedure, other LED packages (e.g., CsPbBr$_3$–SiO$_2$–HF, Sr$_2$SiO$_4$:Eu$^{2+}$ green phosphor, KSF red phosphor, colloidal CsPbBr$_3$ NCs, and CdSe/CdS/ZnS NCs) were also obtained.

**PLQYs measurements**. The absolute PLQYs were calculated by using a fluorescence spectrometer (HAAS-2000) with an integrated sphere excited at a wavelength of 395 nm using a LED chip source.

**Characterization**. The powder X-ray diffraction (XRD) patterns of CsPbBr$_3$–SiO$_2$ and CsPbBr$_3$ NCs were performed by a Bruker D8 Advance X-ray Diffractometer at 40 kV and 30 mA using Cu K$_\alpha$ radiation (λ = 1.5406 Å). The morphologies and elemental distributions and high-angle annular dark−field scanning transmission electron microscopy (HAADF−STEM) images were analyzed by Mira3/MIRA3 (SEM) field emission scanning electron microscope (FESEM) and FEI (TALOS F200X) transmission electron microscope (TEM) instruments. PL emission spectra of the samples were recorded on an Ocean Optics LS-450 spectrometer and a fluorescence spectrometer (HAAS-2000). N$_2$ adsorption-desorption experiments were undertaken isothermally at 77 K on QUADRASORB SI. The photostability measurements of the samples were performed in a temperature and humidity chamber using a fluorescence spectrometer (HAAS-2000). UV-Vis absorption spectra were measured by a UV-Vis spectrophotometer (PerkinElmer Lambda 950). The chemical compositions were determined by X-Ray fluorescence spectroscopy (PANalytical Epsilon 3x). The PL decay curves were recorded on an Edinburgh FLS1000 spectrophotometer with the excitation wavelength at 365 nm.

## Data availability

All relevant data supporting the findings of this study are available from the corresponding authors on request.

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

## Acknowledgements
This study was supported by the National Key R&D Program of China (No. 2018YFC1800600), National Natural Science Foundation of China (NSFC 21773155), and Shanghai Sailing Program (19YF1422200).

## Author contributions
Q.Z. and L.L. proposed the original research idea. The paper was co-written by L.L. and Q.Z. Experiments, including $CsPbBr_3$–$SiO_2$ synthesis, HF etching, LED encapsulation and stability test were performed by Q.Z., B.W., W.Z., L.K., and Q.W. Q.Z., C.Z., Z.L., X.C., and M.L. carried out characterizations and analyses including transmission electron microscope, scanning electron microscopy, powder X-ray diffraction, UV-vis absorption, PL, PL life-time, X-Ray fluorescence spectroscopy, $N_2$ adsorption-desorption and PLQY. All authors discussed the results, interpreted the findings, and reviewed the paper.

## Competing interests
The authors declare no competing interests.
