## [Peer Review File · Nature Communications]

Reviewers' Comments:

Reviewer #1:

Remarks to the Author:

Surface encapsulation is an important tool for maintaining the excellent properties of CsPbBr₃ NCs. The authors reported a strategy to synthesize ceramic-like stable and highly luminescent CsPbBr₃ NCs based on the collapse of the silicon molecular sieve (MS) template at high temperatures. While this study is interesting, the discussion and data analysis need to be improved. This work might be suitable for publication in Nature Communications if the authors can address the following concerns properly:

1). The authors highlighted the obtained CsPbBr₃-SiO₂ powders can maintain their relative photoluminescence (PL) value under illumination and other harsh conditions (Fig.3b, Fig.5b-5c, Fig.6). However, the relative PL intensity cannot accurately and comprehensively evaluate the luminescence performance of solid materials. As this is the key point of this manuscript as written, more professional evaluation methods should be provided to replace the relative PL intensity: a). PLQY for Fig.3b and Fig.5b-5c. b). the luminous efficiency for Fig.6. LED. For the phosphor-LED package to be commercially viable, a luminous efficiency of 100 lm W⁻¹ must be achieved. (Phys. Status Solidi A 2008, 205, 1086–1092; Proc. Natl. Acad. Sci. U.S.A. 2011, 108, 10072–10077; Narukawa, Y.; Ichikawa, M.; Sanga, D.; Sano, M.; Mukai, T.J. Phys. D: Appl. Phys. 2010, 43.).

2). Based on the authors' statement of "A big challenge for CsPbX₃ NCs is that they cannot withstand such a high temperature. In our previous report, the annealing temperature of CsPbBr₃/SiO₂/Al₂O₃ could not exceed 150 °C due to the severe surface oxidations or fusing of CsPbX₃ NCs." However, in this work, the authors can collapse CsPbBr₃ NCs by the silicon molecular sieve (MS) template at very high temperatures (600-900°C). Therefore, it is not entirely clear for me why the CsPbBr₃ is stable at high temperature in this research. I suggest the authors provide more detailed description/discussion, specifically the difference for stability of CsPbX₃ NCs between these two kinds of conditions (< 150 °C for CsPbBr₃/SiO₂/Al₂O₃ vs. 600-900°C in this study) .

3). More detailed synthesis description of CsPbBr₃-SiO₂ powders should be provided. The author described that the collected mixture was ground and calcined at the set temperature for 0.5 h with a heating rate of 5 °C/min in the muffle furnace. What's the atmosphere of the reaction, air or inert gases?

4). The authors found that the CsPbBr₃-SiO₂-HF showed an absorption at 507 nm, which is the same as CsPbBr₃-SiO₂-700 (507 nm). However, their PL peak position has clear blue shifted 2–3 nm (Fig. 4). The authors explained the observations due to the removal of the larger partially-embedded CsPbBr₃ NCs with HF etching. In this case, both absorption and PL should have the ~same blue shift. The authors need to provide more evidence of the conclusion and proper explanation.

Reviewer #2:

Remarks to the Author:

In this paper, silicon molecular sieve was selected to coat and protect CsPbBr₃. Because of silicon's good ability to isolate water and oxygen, CsPbBr₃-SiO₂ can maintain its good optical properties even it was kept in aqueous solution for 50 days. Light stability of CsPbBr₃-SiO₂ has been significantly improved, because no obvious attenuation occurs in the blue light irradiation for 1000 hours. In addition, aging tests at high humidity and high temperature also showed significant improvement in stability. Some comments and questions may help to improve the manuscript, as shown below.

1. In the author's previous work, the annealing temperature should not exceed 150 degrees Celsius, because of severe surface oxidation or fusing of CsPbX₃. But why not now? Just because of the shell material? Can the authors give more details about why CsPbBr₃ can withstand temperatures of 900 °C ?
2. The author used hydrochloric acid solution when testing the water stability, and believed that hydrogen ions could not pass through the SiO₂ shell, showing stable and reliable protection of silica. But there are also hydrogen ions in the water, and the silica itself doesn't react with hydrochloric acid. Is hydrochloric acid really necessary? Does the author want to demonstrate that no anion exchange occurs and that silica is beneficial in preventing anion exchange?
3. If the author wants to show how useful the SiO₂ shell is, it would be better to show a table and give dates about the improved stability in other researches studies by using different coating materials and methods.
4. According to the article, the relative PL strength of 800 and 900 °C is lower than that of 700 °C, because of the concentration? Does PLQY have the same trend? How about stability of CsPbBr₃-SiO₂?
5. The author said that the PL of CsPbBr₃-SiO₂-HF shows a brighter green fluorescence due to the removal of surface partially-embedded CsPbBr₃ NCs, meantime the surface area of CsPbBr₃-SiO₂-HF increased from 9.6 m²/g to 16.7 m²/g. Does the author think that partially coated perovskite has a low PLQY? Is it possible that the real reason is the coating thickness is reduced and less light is absorbed by shell? Does the surface area have an impact on green fluorescence.
6. Why the fluorescence intensity increased in both (a) and (b) in FIG. 6?
7. In SI F4, the particle size increases when temperature increases. Is this because of the sintering temperature?
8. Please explain why the fluorescence intensity of washed CsPbBr₃-SiO₂-700 is higher than that of unwashed CsPbBr₃-SiO₂-700 in SI F5?

Reviewer #3:

Remarks to the Author:

Perovskite NCs are well known for their instability in external environment, which has become the major hurdle to their practical applications. This work embraces this challenge by creatively proposing a robust strategy in breaking the traditional cognition of the instability of perovskite NCs, to synthesize ceramic-like stable and highly luminescent CsPbBr₃ NCs encapsulated in molecular sieve at high temperature up to 700 °C. Surprisingly, the high temperature encapsulation strategy significantly improved the operational stability of perovskite NCs, which were even more stable than the commercial ceramic green phosphor and this strategy could underpin the commercial application of perovskite NCs.

The manuscript is clearly written and sufficient methodological details are provided to ensure that the experiments could be reproduced. The current findings illustrate a promising strategy towards fabrication of highly stable and luminescent ceramic-like inorganic perovskite phosphors and the results demonstrated are novel, robust and convincing (through a broad scope of comprehensive characterizations) hence suitable for publication in Nature Communications after a minor revision.

Main points:

- i) In order to improve the quality of the work and to attract attention from a broader readership, I strongly recommend an additional characterization of the PL properties of ceramic-like CsPbBr₃-SiO₂ synthesized at different temperatures, such as photoluminescent life time. This will enable a more solid basement for the comparison of their luminescent properties.
- ii) The CsPbBr₃-SiO₂ prepared using a mass ratio of 1 (CsPbBr₃):3 (MS) shows the highest luminescent intensity. But why the mass ratio profoundly affects the luminescent properties of ceramic

CsPbBr₃-SiO₂ phosphors is not yet discussed and explained in the manuscript. Is this in relation to the availability of the pores/cavities of silicon molecular sieve that may encapsulate CsPbBr₃?

iii) Authors claim that etching CsPbBr₃-SiO₂ with HF solution can remove incomplete encapsulated CsPbBr₃ NCs on the surface and improve the PLQY. So I want to know if there is any difference in fluorescence lifetime between CsPbBr₃-SiO₂ and CsPbBr₃-SiO₂-HF, which can prove that improved PLQY come from CsPbBr₃-NCs or from thinning of SiO₂ shell?

Minor points:

i) The stability tests mentioned in the manuscript all involved commercial Sr₂SiO₄:Eu²⁺ green phosphor, KSF red phosphor, colloidal CsPbBr₃ NCs and CdSe/CdS/ZnS NCs. So authors should supply the optical characteristics of the luminescent materials, such as PL spectra.

iii) In the manuscript, some descriptions are confusing. For example, original MS and pure MS mean the same material, but they may confuse the readers. Please check the full manuscript and correct them.

ii) The abbreviation of LEDs firstly appears in the Abstract without defining it. This also applies to MS (as the MS template).

iii) As seen from the XRD data of CsPbBr₃-SiO₂-400 sample, the diffraction peaks corresponding to CsPbBr₃ already exist (Figure 1c) although their intensities are not much stronger than those of CsPb₂Br₅. The authors need to clarify if CsPbBr₃ started to form at this calcination temperature or it is already crystallized even when the precursor salt solution was dried at 80 °C.

iv) ... matrix materials are stable and completely cover CsPbBr₃ NCs.... This sentence needs to be corrected.

v) Caption of Figure 1, ... at high temperature...at high temperatures; ...temperature reached to 700 °C ♦ temperature reached 700 °C;

vi) Standard XRD pattern of # 47-1712 was not shown in Figure 2g even this was labeled in this this figure.

vii) The description of "removal of larger partially-embedded CsPbBr₃ nanocrystals with HF etching, which had lower PLQYs" is confusing and needs to be rephrased. I think the authors mean larger partially-embedded CsPbBr₃ nanocrystals had lower PLQYs.

Response to reviewers' comments

To reviewer #1:

1. The authors highlighted the obtained CsPbBr₃-SiO₂ powders can maintain their relative photoluminescence (PL) value under illumination and other harsh conditions (Fig.3b, Fig.5b-5c, Fig.6). However, the relative PL intensity cannot accurately and comprehensively evaluate the luminescence performance of solid materials. As this is the key point of this manuscript as written, more professional evaluation methods should be provided to replace the relative PL intensity: a). PLQY for Fig.3b and Fig.5b-5c. b). the luminous efficiency for Fig.6. LED. For the phosphor-LED package to be commercially viable, a luminous efficiency of 100 lm W⁻¹ must be achieved. (Phys. Status Solidi A 2008, 205, 1086–1092; Proc. Natl. Acad. Sci. U.S.A. 2011, 108, 10072–10077; Narukawa, Y.; Ichikawa, M.; Sanga, D.; Sano, M.; Mukai, T.J. Phys. D: Appl. Phys. 2010, 43.).

Response:

The authors sincerely thank for this suggestion.

a) According to your advices, we have replaced the relative PL intensity data with PLQY for **Figure 3b** and **Figure 5b-5c** in the revised manuscript.

Figure.3 Optical properties of CsPbBr₃-SiO₂. **a** Photoluminescence emission spectra of CsPbBr₃-SiO₂-700 and ceramic Sr₂SiO₄: Eu²⁺ green phosphor, excitation wavelength is 455 nm. **b** Absolute PLQYs of CsPbBr₃-SiO₂ (mass ratio of CsPbBr₃: MS is 1:3, red mark) synthesized at different temperatures, absolute PLQYs of CsPbBr₃-SiO₂ synthesized at 700 °C with different mass ratios of CsPbBr₃: MS (blue mark).

Figure.5 Water resistance and acid resistance of CsPbBr₃-SiO₂. Photographs (a) and relative PLQYs (b) of the CsPbBr₃-SiO₂-700, CsPbBr₃-SiO₂-HF, ceramic Sr₂SiO₄:Eu²⁺ green phosphor, KSF red phosphor, and colloidal CsPbBr₃ NCs after immersed in water for various times. c Relative PLQYs of the CsPbBr₃-SiO₂-700 and CsPbBr₃-SiO₂-HF after immersed in various solvents for 50 days, extra light source: a 450 nm LED light (175mW/cm²).

b) For the question of the luminous efficiency for Fig.6 LED, we agree that aging test by luminous efficiency is a good way to evaluate the stability of the LED package. In this work, the main purpose of Figure 6 was to verify the photostability of CsPbBr₃-SiO₂. The blue-LED chips were just used to provide a strong and identical blue light source to compare the photostability of CsPbBr₃-SiO₂ with other types of phosphors. Actually, it may be not suitable to use luminous efficiency to compare photostabilities for the down conversion materials with different emission wavelengths (CsPbBr₃-SiO₂, colloidal CsPbBr₃ NCs, CdSe/CdS/ZnS NCs, ceramic Sr₂SiO₄:Eu²⁺ green phosphor, and commercial KSF red phosphor). The luminous efficiency was a number related to the ability of the light source to convert energy into electromagnetic radiation and the sensitivity of human eyes, which derived from all the lights of the device including the lights from blue chips, and the phosphors. Therefore, the variation of luminous efficiency during the photostability test is not the direct response from the decay of phosphors self, and using luminous efficiency is not a direct way to evaluate the photostability of phosphors. To better compare the photostabilities of different materials, we still use the PL intensity as the factor but with the

identical blue light source.

Of course, in order to achieve commercial applications, a luminous efficiency of the phosphor-LED package must reach 100 lm/W. Therefore, we packaged the green light-emitting LED device, and the luminous efficiency can reach **111.8 lm/W** as shown in **Figure R1.a-b**. Meanwhile, we also preliminarily explored the potential application of CsPbBr₃-SiO₂ in the field of white LEDs, and the obtained white LEDs by combining CsPbBr₃-SiO₂ and the commercial K₂SiF₆:Mn⁴⁺ red phosphor with a blue LED chip demonstrated a high luminous efficiency of 84 lm/W and a wide color gamut (~125% National Television System Committee standard (NTSC)), which thus promised high brightness and energy saving (**Figure R1.c-d**). Luminous efficiency is limited to less than 100 lm/W, mainly due to the characteristics of CsPbBr₃ NCs, such as a much narrower emission with a full width at half maximum compared with phosphors, and emission peak location (< 525 nm). To our knowledge, the obtained luminous efficiency of the white LEDs is much better than the reported results of all-inorganic-perovskite-based white LEDs as shown in **Table R1**.

Therefore, we not only propose a facile strategy to synthesize ceramic-like stable CsPbBr₃ NCs with excellent optical properties, but also provide the possibility for wider fields, such as backlight display, lighting, and bioimaging.

Figure R1: a-b) Electroluminescence spectra and photograph of CsPbBr₃-SiO₂ sealed with Norland-61 on a blue LED chip (20mA, 2.7V), c-d) Electroluminescence spectra and CIE 1931 color coordinates of as fabricated white LEDs by combining CsPbBr₃-SiO₂ and the commercial

$K_2SiF_6:Mn^{4+}$ red phosphor with a blue LED chip (20mA, 2.7V).

Table R1. Summary of various all-inorganic-perovskite-based white LEDs

Color converter	Operation current [mA]	x/y CIE	LE (lm/W)	NTSC [%]	Ref.
CsPbBr ₃ /SiO ₂ + KSF	20	0.27, 0.30	84	125	This work
PMAO-coated CsPbBr ₃ + Red phosphor	20	0.39, 0.33	56.6	—	1
CsPbBr ₃ /ZrO ₂ + CdSe	1	0.27, 0.30	55	—	2
CsPbBr ₃ + CsPbMnX ₃ @SiO ₂	10	—	68.4	—	3
CsPbBr ₃ NCs + KSF	50	0.34, 0.31	46	118	4
CsPbBr ₃ NCs + CdS	3	0.30, 0.32	51	135	5
CsPbBr ₃ /SiO ₂ NCs + CdSe	6	0.32, 0.30	63.5	—	6
Cs ₄ PbBr ₆ /CsPbBr ₃ + KSF	5	—	73.8	—	7
CaF ₂ -CsPbBr ₃ + CaF ₂ -CsPbBr _{1.2} I _{1.8}	20	0.358, 0.276	62.7	130	8
CsPbBr ₃ NCs + CdSe@ZnS	—	0.32, 0.34	21.6	116	9
CsPbBr ₃ /SiO ₂ + CsPb(Br/I) ₃ /SiO ₂	20	0.33, 0.33	61.2	120	10
CsPbBr ₃ -Zeolite + CsPb(Br _{0.4} I _{0.6}) ₃	20	0.38, 0.37	3.5	114	11
SiO ₂ -CsPbBr ₃ + CsPb(Br _{0.6} I _{0.4}) ₃	—	0.24, 0.28	30	113	12
CsPbBr ₃ /POSS + CsPb(Br/I) ₃ /POSS	20	0.349, 0.383	14.1	—	13
CsPbBr ₃ /PMMA + CdSe/ZnS/PMMA	20	0.31, 0.32	65	125	14
CsPbBr ₃ /TDPA+KSF	20	0.31, 0.29	63	122	15
CsPbBr ₃ QD glass + CaAlSiN ₃ :Eu ²⁺	20	0.40, 0.38	50.5	—	16
CsPbBr ₃ /SiO ₂ + CsPbBr _{1.2} I _{1.8}	20	0.33, 0.36	35.4	127	17
MP-CsPbBr ₃ -SDDA@PMMA/KSF	—	0.271, 0.232	37	102	18
Ethyl cellulose CsPbBr ₃ +Sr ₂ Si ₅ N ₈ :Eu ²⁺	20	—	68	—	19
CsPbBr ₃ + CsPb(Br/I) ₃	20	0.34, 0.48	50-60	—	20
CsPbBr ₃ + KSF	—	0.32, 0.30	98	130	21

1. Chem. Mater. 2019, 31, 1936-1940
2. ACS Nano. 2019, 13, 5366-5374
3. Small. 2019, 15, 1900484.
4. Chem. Eng. J. 2019, 364, 20-27
5. Nano Energy. 2018, 53, 559-566.
6. Nanoscale. 2018, 10, 20131-20139
7. ACS Appl. Mater. Interfaces. 2018, 10, 15905-15912
8. Advanced Optical Materials. 2018, 6, 1701343
9. Advanced Science. DOI: 10.1002/advs.201700335.
10. Advanced Materials. 2016, 28, 10088-10094.

11. *Advanced Functional Materials*. 2017, 27, 1704371.
12. *Angew. Chem., Int. Ed.* 2016, 55, 7924
13. *Chem. Sci.* 2016, 7, 5699
14. *J. Mater. Chem. C*. 2017, 5, 10947.
15. *Nanoscale*. 2017, 9, 15286.
16. *Chem. Commun.* 2017, 53, 11068
17. *Dyes Pigm.* 2018, 149, 246
18. *Chem. Mater.* 2016, 28, 8493–8497
19. *Nanoscale*. 2016, 8, 19523-19526
20. *Advanced Optical Materials*, 2019, 7, 1801663.
21. *Chem. Mater.* 2018, 30, 8546–8554

2. Based on the authors' statement of "A big challenge for CsPbX₃ NCs is that they cannot withstand such a high temperature. In our previous report, the annealing temperature of CsPbBr₃/SiO₂/Al₂O₃ could not exceed 150 °C due to the severe surface oxidations or fusing of CsPbX₃ NCs." However, in this work, the authors can collapse CsPbBr₃ NCs by the silicon molecular sieve (MS) template at very high temperatures (600-900°C). Therefore, it is not entirely clear for me why the CsPbBr₃ is stable at high temperature in this research. I suggest the authors provide more detailed description/discussion, specifically the difference for stability of CsPbX₃ NCs between these two kinds of conditions (< 150 °C for CsPbBr₃/SiO₂/Al₂O₃ vs. 600-900°C in this study).

Response:

Thank you for your advice. According to your suggestions, we have added more detailed discussion and description in the revised manuscript. In our previous report (*Angew. Chem. Int. Ed.* 2017, 56, 8134–8138), CsPbBr₃ NCs were synthesized by the traditional colloidal-synthesis strategy (hot-injection), which offered a kinetically controlled chemical reaction with homogeneous nucleation and growth **with the help of organic solvents and organic ligands**. Then CsPbBr₃ NCs were incorporated into SiO₂/Al₂O₃ by sol-gel reaction of an Al-Si single precursor. It is generally accepted that cross linking can increase the densification of NCs-SiO₂/Al₂O₃ by thermal anneal treatment, which can enhance the photostability. There are two

possible reasons why they cannot withstand high temperature. Firstly, when the annealing temperature (in the air) was too high (exceed 150 °C), the organic ligands on surface of NCs could be oxidized, and also higher temperature may damage or peel off the organic ligands, then CsPbBr₃ NCs will be agglomerated. These factors lead to the fluorescence quenching. Therefore, colloidal synthesized CsPbBr₃ NCs with the existence of organic ligands cannot achieve high temperature annealing. Secondly, the SiO₂/Al₂O₃ sol formed at room temperature was not a dense solid, there was a lot of pores or channels for the ion migration of inter-particles. This ion migration was accelerated with increasing temperature, which unavoidably caused the ripening and fusing of CsPbBr₃ NCs to form larger particles, leading to the fluorescence quenching.

In this work, CsPbBr₃-SiO₂ powders were synthesized by template confined solid-state synthesis at high temperatures (600-900 °C). The growth mechanism of CsPbBr₃ NCs was that the precursors (CsBr, PbBr₂, and bulk CsPbBr₃ crystals) firstly started melting and sublimating at high temperature, then filled into the pores of molecular sieve (MS), and formed CsPbBr₃ NCs when the temperature was cooling down. Simultaneously, the pores of MS started to collapse and finally fused into a dense SiO₂ solid at high temperature. **In particular, organic solvents and organic ligands were not used in this work.** On the other hand, CsPbBr₃ are ionic crystals, which can withstand high temperature annealing. The collapsed MS confined the growth of CsPbBr₃ NCs, more importantly, the seal structure significantly blocked the ion migration between CsPbBr₃ NCs at high temperature. Therefore, CsPbBr₃ NCs confined in collapsed MS were stable at high temperature in this work.

3. More detailed synthesis description of CsPbBr₃-SiO₂ powders should be provided. The author described that the collected mixture was ground and calcined at the set temperature for 0.5 h with a heating rate of 5 °C/min in the muffle furnace. What's the atmosphere of the reaction, air or inert gases?

Response:

The authors thank very much for kind advice. According to your advice, the more detailed synthesis description has been added in the revised manuscript as follows: “The collected mixture was ground and calcined at the set temperature for 0.5 h with a heating rate of 5 °C/min in the muffle furnace under air atmosphere.”

4. The authors found that the CsPbBr₃-SiO₂-HF showed an absorption at 507 nm, which is the same as CsPbBr₃-SiO₂-700 (507 nm). However, their PL peak position has clear blue shifted 2–3 nm (Fig. 4). The authors explained the observations due to the removal of the larger partially-embedded CsPbBr₃ NCs with HF etching. In this case, both absorption and PL should have the ~same blue shift. The authors need to provide more evidence of the conclusion and proper explanation.

Response:

Thank you for pointing out this. We retest the samples several times to avoid the errors which might be from the instruments or manual mistakes. As shown in **Figure 4d**, both PL and absorption of CsPbBr₃-SiO₂-HF have the same blue shift owing to the removal of larger partially-embedded CsPbBr₃ NCs with HF etching.

Figure.4 d Photoluminescence emission and UV-Vis absorption spectra (insert) of CsPbBr₃-SiO₂-700 and CsPbBr₃-SiO₂-HF.

To reviewer #2:

1. In the author's previous work, the annealing temperature should not exceed 150 degrees Celsius, because of severe surface oxidation or fusing of CsPbX₃. But why not now? Just because of the shell material? Can the authors give more details about why CsPbBr₃ can withstand temperatures of 900 °C?

Response:

Thank you for your advice. According to your suggestions, we have added more detailed discussion and description in the revised manuscript. In our previous report (Angew. Chem. Int. Ed. 2017, 56, 8134–8138), CsPbBr₃ NCs were synthesized by the traditional colloidal-synthesis strategy (hot-injection), which offered a kinetically controlled chemical reaction with homogeneous nucleation and growth **with the help of organic solvents and organic ligands**. Then CsPbBr₃ NCs were incorporated into SiO₂/Al₂O₃ by sol-gel reaction of an Al-Si single precursor. It is generally accepted that cross linking can increase the densification of NCs-SiO₂/Al₂O₃ by thermal anneal treatment, which can enhance the photostability. There are two possible reasons why they cannot withstand high temperature. Firstly, when the annealing temperature (in the air) was too high (exceed 150 °C), the organic ligands on surface of NCs could be oxidized, and also higher temperature may damage or peel off the organic ligands, then CsPbBr₃ NCs will be agglomerated. These factors lead to the fluorescence quenching. Therefore, colloidal synthesized CsPbBr₃ NCs with the existence of organic ligands cannot achieve high temperature annealing. Secondly, the SiO₂/Al₂O₃ sol formed at room temperature was not a dense solid, there was a lot of pores or channels for the ion migration of inter-particles. This ion migration was accelerated with increasing temperature, which unavoidably caused the ripening and fusing of CsPbBr₃ NCs to form larger particles, leading to the fluorescence quenching.

In this work, CsPbBr₃-SiO₂ powders were synthesized by template confined solid-state synthesis

at high temperatures (600-900 °C). The growth mechanism of CsPbBr₃ NCs was that the precursors (CsBr, PbBr₂, and bulk CsPbBr₃ crystals) firstly started melting and sublimating at high temperature, then filled into the pores of molecular sieve (MS), and formed CsPbBr₃ NCs when the temperature was cooling down. Simultaneously, the pores of MS started to collapse and finally fused into a dense SiO₂ solid at high temperature. **In particular, organic solvents and organic ligands were not used in this work.** On the other hand, CsPbBr₃ are ionic crystals, which can withstand high temperature annealing. The collapsed MS confined the growth of CsPbBr₃ NCs, but more importantly, the seal structure significantly blocked the ion migration between CsPbBr₃ NCs at high temperature. Therefore, CsPbBr₃ NCs confined in collapsed MS can withstand temperatures of 900 °C in this work.

2. The author used hydrochloric acid solution when testing the water stability, and believed that hydrogen ions could not pass through the SiO₂ shell, showing stable and reliable protection of silica. But there are also hydrogen ions in the water, and the silica itself doesn't react with hydrochloric acid. Is hydrochloric acid really necessary? Does the author want to demonstrate that no anion exchange occurs and that silica is beneficial in preventing anion exchange?

Response:

The authors are very grateful for the kind advice. As the reviewer mentioned, the purpose of hydrochloric acid test was to prove that silica can effectively prevent anion exchange. Hydrochloric acid solution is a harsh chemical environment, containing a large number of H⁺ and Cl⁻ ions. CsPbBr₃-SiO₂ could even survive in the strong acid aqueous solution (1M HCl) for 50 days without obvious PL changes, indicating that Cl⁻ ions cannot pass through the dense SiO₂ shell and cause anion exchange, otherwise the PL peak position of CsPbBr₃ NCs would blue shift (Cl⁻ exchange with Br⁻) as shown in **Figure R2**.

Figure R2: PL emission spectra of CsPbBr₃-SiO₂-700 (a), CsPbBr₃-SiO₂-HF (b), and colloidal

CsPbBr₃ NCs (c) immersed in 1M HCl for various time.

3. If the author wants to show how useful the SiO₂ shell is, it would be better to show a table and give dates about the improved stability in other researches studies by using different coating materials and methods.

Response:

The authors sincerely thank for this insightful advice. According to reviewer' advice, a table about the improved stability by using different coating materials and methods was added in the revised manuscript (**Supplementary Table 7**). To our knowledge, the stabilities (such as, water stability, chemical stability, and photostability) of CsPbBr₃-SiO₂ are much superior to the reported results of conventional perovskite composites obtained by using different coating materials and methods.

Supplementary Table 7. Stability of perovskite composites by using different coating materials and methods.

Composites	Synthetic method	PL QYs	Stability: Remnant PL	Ref.
CsPbBr₃/SiO₂	High temperature encapsulation, MS collapse	71%	Above 100% (1000h, illuminated with 450 nm LED light)	This work
CsPbBr ₃ /SiO ₂	Hot-injection, TMOS hydrolysis	65%	55% (48h, under 100% RH at room temperature)	3
CsPbX ₃ /SiO ₂	Hot-injection, TEOS hydrolysis	84%	40% (4h, water)	4
CsPbBr ₃ /SiO ₂	Hot-injection, TMOS hydrolysis	80%	98% (10h, UV)	5
CsPbBr ₃ /SiO ₂ /PM	Hot-injection, 3-APTES hydrolysis	65%	50% (60 d, stored under ambient conditions)	6
MAPbBr ₃ /SiO ₂	Hot-injection, TMOS hydrolysis	89%	61% (49h, illuminated with 450 nm LED light)	7
CsPbBr ₃ /AlO _x	Hot-injection, Atomic Layer Deposition	56%	50% (8h, 100 mW/cm ² solar irradiation)	8
CsPbBr ₃ -TDPA	Hot-injection	68%	80% (300min, water)	9
CsPbBr ₃ /PS	Electrospinning technique	48%	70% (192h, water)	10
CsPbX ₃ /PMMA	A microfluidic spinning technique	45%	75% (3 d, 30 °C air with 70% humidity))	11
CsPbX ₃ /CaF ₂	Hot-injection	82%	60% ((2 d, air with 100% humidity)	12

MAPbBr ₃ /NaNO ₃	Reprecipitation synthesis	42%	80% (14h, UV)	13
CsPbBr ₃ /NH ₄ Br	Hot-injection, ion-exchange	64%	40% (3.5h, water)	14

List of acronyms and abbreviations

MS: molecular sieve

PM: polystyrene microspheres

TDPA: alkyl phosphate

PS: Polystyrene

PMMA: Polymethyl methacrylate

TMOS: Tetramethyl orthosilicate

TEOS: Tetraethyl orthosilicate

3-APTES: (3-amino-propyl)triethoxysilane

4. According to the article, the relative PL strength of 800 and 900 °C is lower than that of 700 °C, because of the concentration? Does PLQY have the same trend? How about stability of CsPbBr₃-SiO₂?

Response:

The authors are very grateful for reviewer' question.

1) By optimizing reaction temperature, CsPbBr₃-SiO₂-700 exhibited the highest PL intensity and absolute PLQY of 63%. The PLQYs of CsPbBr₃-SiO₂ (800 and 900 °C) were lower than that of CsPbBr₃-SiO₂-700, which had the same trend as relative PL intensities. The concentrations of CsPbBr₃ NCs synthesized at 800 and 900 °C were indeed lower than that at 700 °C. Molecular sieve collapsed completely at 700 °C. Upon increasing the calcination temperature to 800 and 900 °C, the CsPbBr₃ NCs will continue to volatilize/sublimate, leading to lower concentrations. From **Figure 1b**, colors of CsPbBr₃-SiO₂ gradually changed from white to yellow with the synthesis temperature increasing from 400 °C to 700 °C, and then turned into lighter colors when the temperature raised to 800 °C and 900 °C. Meanwhile, the Pb contents in CsPbBr₃-SiO₂ obtained at different calcination temperatures were quantitatively analyzed by inductively coupled plasma optical emission spectroscopy (ICP-OES) as shown in **Figure R3**, which proved that the Pb contents decreased when the temperature reached 800 and 900 °C, meaning lower concentrations

of CsPbBr₃ NCs at 800 and 900 °C. But we think main reason for the lower PLQYs of samples at 800 and 900 °C was uncontrolled aggregation and growth of CsPbBr₃ NCs (**Figure 2** in the manuscript) at high temperature. The growth of larger CsPbBr₃ NCs under high temperature (800 and 900 °C) could be attributed to the softening and collapse of molecular sieve that lead to a weaker template confinement effect. The large sizes of CsPbBr₃ NCs (CsPbBr₃-SiO₂-800, d=20.9 nm; CsPbBr₃-SiO₂-900, d=30.1 nm) have greatly exceeded the exciton Bohr diameter and cause weak quantum confinement effect, which lead to the lower PLQYs.

Figure R3: Pb contents in CsPbBr₃-SiO₂ (Calcination temperature: from 400-900 °C; mass ratio of CsPbBr₃: MS is 1:3).

2) Molecular sieves (MS) collapsed completely at higher calcination temperature (≥ 700 °C). So CsPbBr₃-SiO₂ powders (800 °C and 900 °C) can also show an exceptional stability, because CsPbBr₃ NCs were completely encapsulated into the high temperature annealed dense SiO₂ shell. As we can see, no obvious PLQYs decays were observed in CsPbBr₃-SiO₂ (700 °C, 800 °C and 900 °C) immersed in water for 30 days, indicating that dense SiO₂ shell can be formed at high calcination temperature ≥ 700 °C and provide effective protection for CsPbBr₃ NCs. (**Supplementary Figure 7**).

Supplementary Figure 7. Relative PLQYs and photographs (insert) of CsPbBr₃-SiO₂ (Mass ratio of CsPbBr₃: MS is 1:3) at different calcination temperatures immersed in water for various times.

5. The author said that the PL of CsPbBr₃-SiO₂-HF shows a brighter green fluorescence due to the removal of surface partially-embedded CsPbBr₃ NCs, meantime the surface area of CsPbBr₃-SiO₂-HF increased from 9.6 m²/g to 16.7 m²/g. Does the author think that partially coated perovskite has a low PLQY? Is it possible that the real reason is the coating thickness is reduced and less light is absorbed by shell? Does the surface area have an impact on green fluorescence.

Response:

Thank you for your suggestion.

1) First of all, we cannot confirm that partially-embedded CsPbBr₃ NCs have low PLQY based on the existing data. From **Figure 4d**, PL peak position and UV-Vis absorption edge of CsPbBr₃-SiO₂-HF blue shifted around 2 nm compared with CsPbBr₃-SiO₂, owing to the removal of the larger partially-embedded CsPbBr₃ NCs with HF etching. This evidence cannot prove that partially-embedded CsPbBr₃ NCs have a low PLQY. In order to make the manuscript more rigorous, we have revised the point that partially-embedded CsPbBr₃ NCs have a low PLQY.

2) We think thick SiO₂ shell may hinder the light absorption and conversion of CsPbBr₃ NCs. So a HF solution was used to etch and reduce the thickness of SiO₂ shell, and inevitably remove incomplete encapsulated CsPbBr₃ NCs on the surface of CsPbBr₃-SiO₂. After HF etching, the surface area of CsPbBr₃-SiO₂-HF increased from 9.6 m²/g to 16.7 m²/g because of the thinning of SiO₂, which is beneficial to light absorption and conversion of CsPbBr₃ NCs. Therefore, the real

reason for improving PLQY is the reduced thickness of SiO₂.

3) The surface area does not have decisive effect on green fluorescence in this work. Here, the increase of surface area is to prove the thinning of SiO₂. On the other hand, different surface areas of CsPbBr₃-SiO₂ at different calcination temperatures do not have an impact on PLQYs as shown in **Figure 2h** and **Figure 3b**.

6. Why the fluorescence intensity increased in both (a) and (b) in FIG. 6?

Response:

The authors are very grateful for reviewer' question. As you mentioned, the photoinduced photoluminescence enhancement happens in CsPbBr₃-SiO₂-HF. The phenomenon is termed "photobrightening". This PL enhancement (photobrightening) is possibly due to the photoactivation of CsPbBr₃ NCs before photodegradation. Photobrightening is a general phenomenon in metal chalcogenide NCs, attributed to many factors including the passivation of surface defects [1-3]. And it is widely agreed upon that photobrightening occurs in perovskite nanocrystals through inhibition of nonradiative trapping of excitonic charge carriers [4,5].

[1] Krivenkov V.; Samokhvalov P.; Zvaigzne M.; Martynov, I.; Chistyakov, A.; Nabiev, I. Ligand-Mediated Photobrightening and Photodarkening of CdSe/ZnS Quantum Dot Ensembles. *J. Phys. Chem. C*. 2018, 122, 15761-15771.

[2] McArthur, E. A.; Morris-Cohen, A. J.; Knowles, K. E.; Weiss, E. A. Charge carrier resolved relaxation of the first excitonic state in CdSe quantum dots probed with near-infrared transient absorption spectroscopy. *J. Phys. Chem. B*. 2010, 114, 14514-14520.

[3] Tice, D. B.; Frederick, M. T.; Chang, R. P. H.; Weiss, E. A. Electron migration limits the rate of photobrightening in thin films of CdSe quantum dots in a dry N₂ (g) atmosphere. *J. Phys. Chem. C*. 2016, 115, 3654-3662.

[4] Lorenzon M, Sortino L, Akkerman Q, et al. Role of nonradiative defects and environmental oxygen on exciton recombination processes in CsPbBr₃ perovskite nanocrystals[J]. *Nano letters*, 2017, 17(6): 3844-3853.

[5] Yuan G, Ritchie C, Ritter M, et al. The degradation and blinking of single CsPbI₃ perovskite quantum dots[J]. *The Journal of Physical Chemistry C*, 2017, 122(25): 13407-13415.

7. In SI F4, the particle size increases when temperature increases. Is this because of the sintering temperature?

Response:

Thanks for your question. We agree that the reason for the particle size increasing with the raising of temperature is the sintering temperature. From **Figure 2** and **Supplementary Figure 4**, with increasing calcination temperature, the average particle sizes of CsPbBr₃ NCs increased gradually. The increased sizes of CsPbBr₃ NCs may be attributed to the softening and collapse of pores of molecular sieve with increasing temperature, leading to a weaker template confinement effect, which allowed to grow larger particles.

8. Please explain why the fluorescence intensity of washed CsPbBr₃-SiO₂-700 is higher than that of unwashed CsPbBr₃-SiO₂-700 in SI F5?

Response:

The authors sincerely thank for this insightful advice. We think the improved PL intensity of the water washed CsPbBr₃-SiO₂-700 could be attributed to the removing of reaction residues and salts from the surface of SiO₂, because the reaction residues and salts (CsBr or PbBr₂) usually were non-luminescent. From **Supplementary Figure 5a**, some reaction residues and salts were observed on the surface of unwashed CsPbBr₃-SiO₂-700. In order to further explain reaction residues and salts after water washing, X-Ray Fluorescence measurement (XRF) was used to confirm the Cs, Pb, and Br element contents of the unwashed and washed CsPbBr₃-SiO₂-700. As shown in **Supplementary Figure 6** in the revised Supplementary Information, after water washing, Cs, Pb, and Br element contents of CsPbBr₃-SiO₂-700 decreased obviously, because of removing of reaction residues and salts from the surface of SiO₂ that could influence the light absorption and conversion of CsPbBr₃ NCs in the SiO₂.

Supplementary Figure 6. Cs, Pb, and Br element contents of the unwashed and water washed CsPbBr₃-SiO₂-700 determined from XRF.

To reviewer #3:

Main points:

1. In order to improve the quality of the work and to attract attention from a broader readership, I strongly recommend an additional characterization of the PL properties of ceramic-like CsPbBr₃-SiO₂ synthesized at different temperatures, such as photoluminescent life time. This will enable a more solid basement for the comparison of their luminescent properties.

Response:

The authors sincerely thank for this insightful advice. According to your advice, photoluminescent lifetimes of CsPbBr₃-SiO₂ synthesized at different temperatures have been provided in the revised manuscript. **Supplementary Figure 8** and **Supplementary Table 4** presented PL decay curves of CsPbBr₃-SiO₂ synthesized at different temperatures. The average lifetimes of CsPbBr₃-SiO₂ gradually increased with the synthesis temperature increasing from 400 °C to 700 °C, and then

began to decrease when the temperature raised to 800 °C and 900 °C. Particularly, the longest average lifetime was 21.81 ns at 700 °C. The longer lifetimes usually indicated the suppression of the nonradiative decay, and the generated excitons were more inclined to recombine with radiative path, which were in consistent with the PLQYs with the change of synthesis temperatures.

Supplementary Figure 8: Time-resolved PL decay spectra of CsPbBr₃-SiO₂ synthesized at different temperatures.

Supplementary Table 4: The fitting parameters of the decay curves for CsPbBr₃-SiO₂ synthesized at different temperatures.

Samples	τ_1 (ns)	A ₁ (%)	τ_2 (ns)	A ₂ (%)	τ_3 (ns)	A ₃ (%)	τ_{ave} (ns)
CsPbBr ₃ -SiO ₂ -400	0.78	95.84	11.36	4.16			4.86
CsPbBr ₃ -SiO ₂ -500	1.70	77.43	6.24	20.81	31.17	1.76	8.67
CsPbBr ₃ -SiO ₂ -600	1.85	31.55	6.03	60.23	19.32	8.22	9.25
CsPbBr ₃ -SiO ₂ -700	2.84	38.70	10.21	49.66	39.25	11.65	21.81
CsPbBr ₃ -SiO ₂ -800	3.58	41.07	11.04	51.47	38.28	7.46	17.72
CsPbBr ₃ -SiO ₂ -900	1.69	43.30	6.52	50.62	24.20	6.08	10.61

2. The CsPbBr₃-SiO₂ prepared using a mass ratio of 1 (CsPbBr₃):3 (MS) shows the highest luminescent intensity. But why the mass ratio profoundly affects the luminescent properties of ceramic CsPbBr₃-SiO₂ phosphors is not yet discussed and explained in the manuscript. Is this in relation to the availability of the pores/cavities of silicon molecular sieve that may encapsulate

CsPbBr₃?

Response:

Thank you for your advice. According to your advice, we further explored the effect of mass ratio (CsBr/PbBr₂: MS) on luminescent properties of CsPbBr₃-SiO₂. More detailed discussion and description were added in the revised Supplementary Information as follows:

Supplementary Figure 9 illustrated the photographs of CsPbBr₃-SiO₂ with different mass ratio of CsBr/PbBr₂: MS synthesized at 700 °C. The colors of CsPbBr₃-SiO₂ gradually changed from white to deeper yellow with the mass ratio of CsBr/PbBr₂: MS increasing from 1:10 to 10:1. After water washing, the colors of CsPbBr₃-SiO₂ (CsBr/PbBr₂: MS=3:1, 5:1, 10:1, means excess of CsBr/PbBr₂) began to lighten, which indicated that a majority of CsPbBr₃ bulk crystals were formed on the outside of MS when employed exceedingly high mass ratio of CsBr/PbBr₂: MS.

Similar results can be obtained from UV-Vis absorption spectra (**Supplementary Figure 10**). From **Supplementary Figure 10a**, absorption intensities of CsPbBr₃-SiO₂ increased gradually with the raising of mass ratio of CsBr/PbBr₂: MS. After water washing, the absorption intensities of CsPbBr₃-SiO₂ with high mass ratio of CsBr/PbBr₂: MS (3:1, 5:1, 10:1) decreased obviously, indicating that the bulk crystals outside of MS were removed by water washing.

The XRD patterns (**Supplementary Figure 11**) confirmed the formation of cubic CsPbBr₃ NCs (PDF# 54-0752) with the mass ratio of CsBr/PbBr₂: MS=1:3, but the characteristic diffraction peaks were not observed in CsPbBr₃-SiO₂ (CsBr/PbBr₂: MS=1:10, 1:5), owing to the lower concentration of CsPbBr₃ NCs. However, except the existence of the cubic CsPbBr₃ NCs, we also observed the sharp diffractions from Cs₄PbBr₆ (PDF# 01-073-2478) in CsPbBr₃-SiO₂ with the increase of mass ratio of CsBr/PbBr₂: MS from 1:1 to 10:1, which indicated that bulk crystals (CsPbBr₃ and Cs₄PbBr₆) were formed.

Therefore, excessive CsBr/PbBr₂ raw materials did not promote PL intensities or PLQYs of resultant CsPbBr₃-SiO₂, and the main reason can be attributed to the availability of the pores/cavities of MS that may encapsulate CsPbBr₃ NCs. By optimizing the mass ratio of CsBr/PbBr₂: MS, CsPbBr₃-SiO₂ (1:3) (t=700 °C, mass ratio of CsPbBr₃: MS is 1:3) exhibited the highest absolute PLQY of 63%.

Supplementary Figure 9: Photographs of the unwashed $\text{CsPbBr}_3\text{-SiO}_2$ powders (upper) and water washed $\text{CsPbBr}_3\text{-SiO}_2$ powders (bottom) with different mass ratio of CsBr/PbBr_2 : MS synthesized at 700°C . 1# CsBr/PbBr_2 : MS=1:10; 2# CsBr/PbBr_2 : MS=1:5; 3# CsBr/PbBr_2 : MS=1:3; 4# CsBr/PbBr_2 : MS=1:1; 5# CsBr/PbBr_2 : MS=3:1; 6# CsBr/PbBr_2 : MS=5:1; 7# CsBr/PbBr_2 : MS=10:1.

Supplementary Figure 10: UV-Vis absorption spectra of (a) unwashed $\text{CsPbBr}_3\text{-SiO}_2$ powders with different mass ratio of CsBr/PbBr_2 : MS synthesized at 700°C , (b) water washed

CsPbBr₃-SiO₂ powders with different mass ratio of CsBr/PbBr₂: MS synthesized at 700 °C, (c-i) UV-Vis absorption spectra of unwashed and washed CsPbBr₃-SiO₂ powders with different mass ratio of CsBr/PbBr₂: MS (from 1:10 to 10:1) synthesized at 700 °C.

Supplementary Figure 11: XRD patterns of CsPbBr₃-SiO₂ powders with different mass ratio of CsBr/PbBr₂: MS at 700 °C. (CsPbBr₃: PDF# 54-0752; Cs₄PbBr₆: PDF# 01-073-2478), 1# CsBr/PbBr₂: MS=1:10; 2# CsBr/PbBr₂: MS=1:5; 3# CsBr/PbBr₂: MS=1:3; 4# CsBr/PbBr₂: MS=1:1; 5# CsBr/PbBr₂: MS=3:1; 6# CsBr/PbBr₂: MS=5:1; 7# CsBr/PbBr₂: MS=10:1.

3. Authors claim that etching CsPbBr₃-SiO₂ with HF solution can remove incomplete encapsulated CsPbBr₃ NCs on the surface and improve the PLQY. So I want to know if there is any difference in fluorescence lifetime between CsPbBr₃-SiO₂ and CsPbBr₃-SiO₂-HF, which can prove that improved PLQY come from CsPbBr₃-NCs or from thinning of SiO₂ shell?

Response:

The authors sincerely thank for this insightful advice. According to your advice, fluorescence lifetimes of CsPbBr₃-SiO₂-700 and CsPbBr₃-SiO₂-HF have been provided in the revised manuscript. There was no significant difference in average fluorescence lifetimes between CsPbBr₃-SiO₂-700 ($\tau=21.81$ ns) and CsPbBr₃-SiO₂-HF ($\tau=22.14$ ns) (**Supplementary Figure 14 and Supplementary Table 6**). Therefore, the HF etching did not change the structures and properties of CsPbBr₃ NCs. In other words, the improved PLQY was mainly attributed to the thinning of SiO₂ shell.

Supplementary Figure 14: Time-resolved PL decays spectra of CsPbBr₃-SiO₂ and CsPbBr₃-SiO₂-HF.

Supplementary Table 6. The fitting parameters of the decay curves for CsPbBr₃-SiO₂-700 and CsPbBr₃-SiO₂-HF.

Samples	τ_1 (ns)	A ₁ (%)	τ_2 (ns)	A ₂ (%)	τ_3 (ns)	A ₃ (%)	τ_{ave} (ns)
CsPbBr ₃ -SiO ₂ -700	2.84	38.70	10.21	49.66	39.25	11.65	21.81
CsPbBr ₃ -SiO ₂ -HF	2.84	38.16	10.08	50.54	40.06	11.30	22.14

Minor points:

1) The stability tests mentioned in the manuscript all involved commercial Sr₂SiO₄:Eu²⁺ green phosphor, KSF red phosphor, colloidal CsPbBr₃ NCs and CdSe/CdS/ZnS NCs. So authors should supply the optical characteristics of the luminescent materials, such as PL spectra.

Response:

Thank you for your advice. According to your advice, the PL spectra of all involved luminescent materials have been added in revised Supplementary Information (**Supplementary Figure 18**).

2) In the manuscript, some descriptions are confusing. For example, original MS and pure MS mean the same material, but they may confuse the readers. Please check the full manuscript and correct them.

Response:

Thank you for your advice. According to your advice, some misleading descriptions have been revised in the revised manuscript.

3) The abbreviation of LEDs firstly appears in the Abstract without defining it. This also applies to MS (as the MS template).

Response:

Thank you for your advice. According to your advice, the abbreviation of LEDs have been defined in the Abstract.

4) As seen from the XRD data of CsPbBr₃-SiO₂-400 sample, the diffraction peaks corresponding to CsPbBr₃ already exist (Figure 1c) although their intensities are not much stronger than those of CsPb₂Br₅. The authors need to clarify if CsPbBr₃ started to form at this calcination temperature or it is already crystallized even when the precursor salt solution was dried at 80 °C.

Response:

The authors thank you very much for the kind advice. From **Figure 1c**, the sharp diffractions of CsPbBr₃ and CsPb₂Br₅ bulk crystals can be observed in CsPbBr₃-SiO₂-400. Meanwhile, CsPbBr₃ and CsPb₂Br₅ bulk crystals can also appeared in the dried raw mixtures without calcination (**Supplementary Figure 1b**), because CsPbBr₃ and CsPb₂Br₅ bulk crystals were crystallized when the precursor salt solutions were dried at 80 °C. So there was no obvious chemical reaction or phase transition at 400 °C.

5) ... matrix materials are stable and completely cover CsPbBr₃ NCs.... This sentence needs to be corrected.

Response:

Thank you for your advice. According to your advice, this sentence have been revised in the revised manuscript.

6) Caption of Figure 1, ... at high temperature...at high temperatures; ...temperature reached to 700 oC ◇ temperature reached 700 oC.

Response:

Thank you for your advice. According to your advice, the corresponding mistakes have been revised in the revised manuscript.

7) Standard XRD pattern of # 47-1712 was not shown in Figure 2g even this was labeled in this figure.

Response:

Thank you for your advice. According to your advice, standard XRD pattern of # 49-1712 have been provided (**Figure 2g**) in the revised manuscript.

8) The description of “removal of larger partially-embedded CsPbBr₃ nanocrystals with HF etching, which had lower PLQYs” is confusing and needs to be rephrased. I think the authors mean larger partially-embedded CsPbBr₃ nanocrystals had lower PLQYs.

Response:

Thank you for your advice. According to your advice, the corresponding description has been revised in the revised manuscript.

Reviewers' Comments:

Reviewer #1:

Remarks to the Author:

All of my comments/concerns were addressed by the authors in the revised manuscript. The work is publishable and further review is not needed.

Reviewer #2:

Remarks to the Author:

Accept

In order to improve the stability of CsPbBr₃ nanocrystals, CsPbBr₃-SiO₂ was encapsulated into silica derived from molecular sieve (MS) at high temperature (600-900°C). CsPbBr₃ NCs have a ceramic-like stability and have good performance under illumination for a long time (1000 h), and can be used in other fields because of their excellent water/acid-resistance. Although there are many problems in their first draft, such as missing a lot of key tests, inadequate data analysis, and insufficient reasons for some views. However, the revised manuscript makes up for the above deficiencies. Researchers in other related disciplines may be attracted by their work. I suggest that it can be published.

Reviewer #3:

Remarks to the Author:

The authors fully address the question that I raised in a thorough and systematic way and I am satisfied with the responses. The revisions strengthen this manuscript by demonstrating a strategy in profoundly improving the stability of ceramic-like CsPbBr₃ nanocrystals and demonstrate the potential of this material in practical applications in a wide scope of applications in LEDs, back-lighting etc. Now the manuscript is publishable in Nature Communications.